# Adiponectin-Based Peptide (ADP355) Inhibits Transforming Growth Factor-β1-Induced Fibrosis in Keloids

**DOI:** 10.3390/ijms21082833

**Published:** 2020-04-18

**Authors:** Claudia C. Darmawan, Sara E. Montenegro, Gwanghyun Jo, Novi Kusumaningrum, Si-Hyung Lee, Jin-Ho Chung, Je-Ho Mun

**Affiliations:** 1Department of Dermatology, Seoul National University College of Medicine, Seoul 03080, Korea; claudiachristin93@yahoo.com (C.C.D.); sarym89@gmail.com (S.E.M.); chrisjoe17@naver.com (G.J.); always5515@gmail.com (S.-H.L.); jhchung@snu.ac.kr (J.-H.C.); 2Institute of Human-Environment Interface Biology, Seoul National University, Seoul 03080, Korea; 3Department of Dermatology and Venereology, Diponegoro University, Semarang 50275, Indonesia; novi.zamrudi@gmail.com

**Keywords:** keloid, adiponectin, peptide, treatment, fibrosis, scarring, ADP355, xenograft

## Abstract

Keloids, benign cutaneous overgrowths of dermal fibroblasts, are caused by pathologic scarring of wounds during healing. Current surgical and therapeutic modalities are unsatisfactory. Although adiponectin has shown an antifibrotic effect, its large size and insolubility limit its potential use in keloid treatment. We investigated the effect of a smaller and more stable adiponectin-based peptide (ADP355) on transforming growth factor β1 (TGF-β1)-induced fibrosis in a primary culture of keloid fibroblasts prepared from clinically obtained keloid samples. Xenograft of keloid tissues on athymic nude mice was used to investigate the effect of intralesional injection of ADP355. ADP355 significantly attenuated the TGF-β1-induced expression of procollagen type 1 in keloid fibroblasts (*p* < 0.05). Moreover, it inhibited the TGF-β1-induced phosphorylation of SMAD3 and ERK, while amplifying the phosphorylation of AMP-activated protein kinase (*p* < 0.05). Knockdown of adiponectin receptor 1 reversed the attenuation of procollagen expression in ADP355-treated TGF-β1-induced fibrosis (*p < 0.05*). ADP355 also significantly reduced the gross weight and procollagen expression of keloid tissues in xenograft mice compared to control animals. These results demonstrate the therapeutic potential of the adiponectin peptide ADP355 for keloids.

## 1. Introduction

Keloids are a type of excessive scarring, which results from aberrations in physiologic wound healing. They generally occur after apparent injury to the skin tissue but may develop over several years after minor injuries and may even form spontaneously without any perceivable injuries. Keloids persist for long periods and do not regress spontaneously [1]. This condition is characterized by benign cutaneous hyperproliferation of dermal fibroblasts; overproduction of collagen, fibronectin and other extracellular matrix components; and increased infiltration of inflammatory cells with high recurrence rates [2,3]. These lesions can significantly affect the patient’s quality of life, both physically and psychologically. Appropriate management of keloids is therefore necessary. However, current therapeutic modalities often exhibit limited efficacies [4].

Adiponectin, an adipokine that is predominantly secreted by adipose tissue, exerts its multifunctional effect through interactions with cell-surface adiponectin receptors (AdipoRs) [5,6]. Adiponectin elicits several downstream signaling events. APPL1 (adaptor protein containing pleckstrin homology domain, phosphotyrosine binding domain, and leucine zipper motif) mediates the intracellular signal transduction of the adiponectin receptor pathway. It activates AMP-activated protein kinases (AMPK), PPAR-α, calcium/calmodulin-dependent protein kinase kinase (CaMKK-β), and p38 MAPK. [7,8]. Adiponectin has distinct target tissue specificity and modulates unique biological processes; it improves insulin sensitivity, regulates energy metabolism, and modulates the inflammatory response [9].

A recent study has shown that adiponectin attenuates connective tissue growth factor-induced keloid fibrosis, migration, and extracellular matrix (ECM) overproduction [10]. These findings indicate a possible novel role of adiponectin in keloid treatment. However, adiponectin is a relatively large (244 amino acids) cytokine [11]. In 2011, an adiponectin-based short peptide, named ADP355 (H-DAsn-Ile-Pro-Nva-Leu-Tyr-DSer-Phe-Ala-DSer-NH_2_), that can mimic adiponectin action was developed as a suitable drug for preclinical and clinical development [5]. Previous studies have shown that adiponectin-based peptide (ADP355) has an antifibrotic effect on liver fibrosis and systemic sclerosis [12,13,14]. In this study, we investigated the effect of ADP355 in keloid fibroblasts and xenograft mice to determine the therapeutic benefits of adiponectin peptide for the treatment of keloids.

## 2. Results

### 2.1. Effect of ADP355 on Keloid Fibroblast Viability and the AMP-Activated Protein Kinase (AMPK) Pathway 

Firstly, we performed a keloid cell viability test to exclude the possible cytotoxic effect of ADP355 that may indirectly affect the production of procollagen type 1. We investigated the effect of ADP355 on keloid cell viability at 5, 10, 50, and 100 µg/mL. ADP355 did not affect keloid fibroblast viability at doses of 5 to 10 µg/mL, although significantly decreased cell viability was observed at doses ranging from 50 to 100 µg/mL (Figure 1A). Based on these results, further experiments were conducted using an ADP355 dose ranging from 5 to 10 µg/mL. At 10 µg/mL, ADP355 increased the phosphorylation of AMPK in a time-dependent manner, with the most significant activity observed 24 h after treatment of the keloid fibroblasts (Figure 1B).

### 2.2. ADP355 Suppressed the Production of Procollagen Type 1 Expression

The effect of ADP355 on procollagen expression was investigated in transforming growth factor β1 (TGF-β1) (5 ng/mL)-induced fibroblasts. TGF-β1 significantly increases procollagen production. However, production was attenuated following treatment with 10 μg/mL of ADP355 to levels comparable to those of the positive control of 10 μg/mL adiponectin recombinant (AdipoQ). These results suggested that 10 µg/mL ADP355 attenuated TGF-β1-induced procollagen type 1 expression in keloid fibroblasts (Figure 2). Subsequent experiments were conducted using a dose of 10 µg/mL ADP355.

### 2.3. ADP355 Attenuated Phosphorylation of ERK and SMAD3 and Accentuated AMPK in TGF-β1-Treated Keloid Fibroblasts

TGF-β-induced cellular responses involve the phosphorylation of several signaling pathways, such as phosphorylation of SMAD2, SMAD3, AMPK, and ERK, which were investigated in this study. TGF-β significantly increased the levels of p-SMAD2, p-SMAD3, p-AMPK, and p-ERK. ADP355 (10 µg/mL) significantly inhibited the TGF-β-induced phosphorylation of SMAD3 and ERK and amplified p-AMPK phosphorylation (Figure 3). However, there was no significant difference observed in the phosphorylation of SMAD2. 

### 2.4. Knockdown of AdipoR1 Attenuated the Inhibitory Effect of ADP355 on TGF-β-Induced Fibrosis

Adiponectin Receptor 1 (AdipoR1) has been shown to substantially contribute to the ADP355-induced antifibrotic effect, as well as adiponectin recombinant-mediated pathways in keloids [5,10]. Therefore, we used siRNA targeting AdipoR1 to determine whether the antifibrotic effect of ADP355 is reversed by the knockdown of AdipoR1. Specific knockdown of AdipoR1 was confirmed through qRT-PCR expression of AdipoR1, relative to negative-control siRNA (Figure 4A). We confirmed that knockdown of AdipoR1 reversed the attenuation of procollagen expression on ADP355 and AdipoQ-treated TGF-β-induced fibrosis (Figure 4B, siRNA+ vs. siRNA-, *p* < 0.05).

### 2.5. Intralesional Injection of ADP355 Reduced the Size and Procollagen Expression of Xenotransplanted Keloid Tissue

To evaluate the effect of ADP355 in keloid animal models, we implanted keloid tissues excised from patients onto the backs of mice. Following six intralesional injections over 2 weeks (thrice/week), the grafts were excised (Figure 5A). The weight of ADP355-treated lesions were significantly lower than those of the vehicle-treated mice (*p* < 0.05, Figure 5B). Western blot analysis showed that procollagen expression was significantly reduced, while expression of p-AMPK was increased, following treatment with ADP355 and AdipoQ (*p* < 0.05, Figure 5C). 

## 3. Discussion

Keloid formation is the consequence of the pathologic wound healing process, which involves complex regulatory pathways [15]. Dysregulation of TGF-β has been shown to play a prominent role in determining the outcome of keloid formation [16,17]. Inhibition of TGF- β/Smad and MAPK/ERK signaling has been shown to counteract TGF-β-induced keloid fibroblast proliferation, migration, and invasion, and to simultaneously reduce collagen production [18].

Adiponectin is an adipokine that is predominantly secreted by the adipose tissue and exerts a multifunctional effect through interaction with cell-surface adiponectin receptors (AdipoRs). It plays a role in multiple organs including the liver, kidney, pancreas, and muscle. Adiponectin reportedly attenuates tissue fibrosis in the liver and kidney [19]. It has also been reported to be involved in the pathogenesis of keloids by influencing cell proliferation and migration and by inducing the overproduction of ECM in keloid fibroblasts [10]. However, its extremely insoluble C-terminal globular domain and large peptide fragments hinder the development of whole adiponectin protein as a drug or for use in other treatment modalities [5]. In this regard, adiponectin-based short peptide agonists may be used as potential alternative therapeutic agents. ADP355 is a 10-amino acid-long peptide that has been reported to have an antitumor effect in breast cancer, an antifibrotic effect on the liver, and to cause partial attenuation of protease inhibitor-induced cognitive impairment and brain injury [5,12,20].

Several studies have demonstrated promising application of synthetic peptides for the inhibition and reduction of scarring [21,22,23]. In a multiple randomized controlled trial, the effect of aCT1, a transmembrane protein Cx43-mimicking peptide, was investigated. The peptide improved overall scar thickness and pigmentation [22]. Most recently, growth hormone-releasing peptide 6 (GHRP6) has been shown to modulate the expression of several proteins that have been implicated in keloid formation [21].

In this experiment, we explored the effect of an adiponectin-based receptor agonist, ADP355, on TGF-β1-induced fibrosis of keloid fibroblasts. As AMPK inhibits TGF-β-induced transcription downstream of SMAD3 COOH-terminal phosphorylation and nuclear translocation [24], its amplification can antagonize TGF-β1-induced collagen production. In this study, both adiponectin and ADP355 significantly upregulated the phosphorylation of AMPK in keloid fibroblasts, which, in turn, resulted in a reduction of TGF-β1-induced procollagen expression. ERK MAP kinases phosphorylate receptor-activated SMADS and regulate their nuclear translocation [25], and the inhibition of TGF-β/Smad and MAPK/ERK signaling pathways can antagonize collagen production [18]. Our results revealed that both adiponectin and ADP355 inhibited the TGF-β1-induced phosphorylation of SMAD3 and ERK. In addition, we showed that knockdown of AdipoR1 reversed the antifibrotic effect of both ADP355 and AdipoQ. Collectively, ADP355 reduced the effect of TGF-β1-induced procollagen type 1 expression by attenuating phosphorylation of SMAD3 and ERK signaling and amplifying the p-AMPK pathway through AdipoR1.

Direct xenografting of keloid tissue into athymic nude mice is one of the most representative approaches for the study of keloid tissue [26]. This approach allows the observation of human tissue in an ex vivo environment by preserving complex human cell–cell interactions, while limiting the impact of environmental factors [26]. Our results show that the intralesional injections of ADP355 significantly reduced procollagen expression and amplified phosphorylation of AMPK. In line with these results, the weight of ADP355-treated keloid tissue reduced significantly in comparison to that of control tissue.

In conclusion, our study demonstrated the antifibrotic effect of ADP355 in keloid fibroblasts and xenografted keloid tissue in mice. These results suggest a possible therapeutic application of ADP355 in keloid treatment. 

## 4. Materials and Methods 

### 4.1. Isolation of Keloid and Keloid Dermal Fibroblast Primary Cell Culture 

This study was approved by the Seoul National University Institutional Review Board, and was conducted according to the principles of the Declaration of Helsinki. Informed consent was obtained from all patients. Keloid tissues were obtained from seven patients during keloid removal surgery. Patients with keloids were all diagnosed by pathological examination. No patients received any treatment six months prior to surgery. Human dermal fibroblasts were isolated by mechanical and enzymatic digestion using a previously described protocol [27]. Cells were cultured in Dulbecco’s modified Eagle’s medium (DMEM) purchased from Welgene (Gyeongsan, South Korea) with penicillin (400 U/mL), streptomycin (50 mg/mL) purchased from Life Technologies (Rockville, MD, USA), and 10% FBS in a humidified 5% CO_2_ atmosphere at 37 °C. Keloid fibroblasts between passage 1 and 4 were used for the subsequent experiments.

### 4.2. Adiponectin Peptide Treatment

ADP355 (H-DAsn-Ile-Pro-Nva-Leu-Tyr-DSer-Phe-Ala-DSer-NH_2_) was purchased from Peptron (Daejeon, South Korea). Cells were seeded on a 6-well plate in duplicate, and upon reaching 80–90% confluence, cells were starved in serum-free medium for 24 h. Cells were treated with TGF-β and ADP355 or AdipoQ simultaneously, and then harvested 24 h after treatment.

### 4.3. Cell Viability Analysis

Cells were seeded in a 96-well plate at a density of 1800 cells per well and incubated for 24 h at 37 °C. Cells were cultured to 60% confluence and starved for another 24 h in serum-free DMEM and treated with ADP355 (5–100 µg/mL) for 24 h. Cell viability was tested using the Ezy-cytox cell viability assay kit (Daeil Bio, Suwon, South Korea) according to the manufacturer’s instructions. Briefly, Ezy-cytox reagent was added to each well and incubated for 1 h. The absorbance of collected medium was determined spectrophotometrically using a microplate reader (Molecular Devices, Sunnyvale, CA, USA) at 450/650 nm.

### 4.4. Western Blot Analysis

Following the treatment under appropriate conditions for 24 h, cells were harvested, washed with ice-cold PBS, and lysed with radioimmunoprecipitation assay lysis buffer (EMD Millipore, Billerica, MA, USA) mixed with a protease inhibitor mixture (Roche Applied Science, Penzberg, Germany) and a phosphatase inhibitor mixture (Sigma-Aldrich; Merck KgaA, St. Louis, MO, USA). Cell lysates were centrifuged at 13,500 *g* at 4 °C for 15 min, and the resultant supernatants were collected. The total extract protein was quantified using a bicinchoninic acid assay reagent (Sigma-Aldrich; Merck KGaA). Equal amounts of protein were then separated using 10% sodium dodecyl sulphate polyacrylamide gel electrophoresis and transferred to polyvinylidene difluoride membranes (Roche Applied Science). The membranes were blocked with 5% skim milk diluted in tris-buffered saline containing 0.1% Tween-20, followed by overnight incubation with the target protein primary antibodies β-actin (Santa Cruz, CA, USA), procollagen type-1 (Developmental studies Hybridoma Bank, IA, USA), p-ERK (Santa Cruz, CA, USA), t-AMPK (Cell Signaling, MA, USA), p-AMPK (Cell Signaling), p-SMAD2 (Santa Cruz), and p-SMAD3 (Santa Cruz). β-actin was used as a loading control. The membranes were washed and incubated with mouse polyclonal antibodies (Genetex, CA, USA) against β-actin and procollagen, and rabbit polyclonal antibodies (Genetex) against p-ERK, p-SMAD2, p-SMAD3, p-AMPK, and t-AMPK. Immunoreactive bands were then visualized using the enhanced chemiluminescence detection system (Thermo Fisher Scientific, Inc., MA, USA). Band density was measured using ImageJ software version 1.51w (National Institutes of Health, Bethesda, MD, USA). The protein expression was normalized to that of β-actin.

### 4.5. siRNA Transfection

AdipoR1 in the keloid dermal fibroblasts was depleted through transfection of 100 nM siRNA into the keloid fibroblasts using lipofectamine 2000 (Invitrogen, Carlsbad, CA, USA) following the manufacturer’s protocol. The existence of siRNA off-target effects was excluded by use of a negative control siRNA. After 18 h of transfection, the cells were treated following the same treatment protocol with/without TGF-β1 and ADP355 or AdipoQ in serum-free DMEM for 24 h. mRNA expression of AdipoR1 was examined by quantitative real-time reverse transcription PCR (RT-PCR). Procollagen expression was examined by western blot analysis using the previously described protocol.

### 4.6. Animal Model and Keloid Xenotransplantation

Keloid tissues were acquired from affected patients during surgical excision. The tissues were cut into small pieces using a 5 mm punch. They were preserved in normal saline and implanted on the backs of female athymic nude BALB/c mice (aged 12 weeks) within 3 h of acquisition. The gross weight of each tissue sample was calculated prior to implantation. The weight of the implanted keloid tissue ranged from 0.07 to 0.1 g (mean, 0.087 g). The mice were anesthetized with isoflurane, and three 5 mm punches were used to make a recipient defect on the back of the mice where the keloids could be transplanted and sutured. Once the wound completely healed and was covered with flesh-colored skin on day 24, the lesions on each mouse were assigned for injection with vehicle (phosphate buffer solution), 10 µM adiponectin-based peptide ADP355, or 10 µM adiponectin recombinant for six treatments over two weeks. Following this, the lesions were excised to determine the effect of the treatment. This study was approved by The Institutional Animal Care and Use Committee (IACUC) of Seoul National University, and the number of animals used were kept to a minimum in order to reduce their suffering, according to protocol.

### 4.7. Tissue Analysis

The tissue samples’ gross weights were evaluated. The weights of the grafts after six injections within two weeks were normalized against the weights of the grafts before implantation (mean ± standard deviation). The tissue weight reduction in treatment groups was compared to that in the control group. The remaining snap-frozen tissues were lysed with radioimmunoprecipitation assay lysis buffer (EMD Millipore, Billerica, MA, USA) mixed with protease inhibitor mixture (Roche Applied Science, Penzberg, Germany), and western blot analysis was carried out according to the protocol previously described.

### 4.8. Statistical Analysis 

The results are represented as mean values (SD), each of which represents at least three independent representative experiments. Mann–Whitney U test was used to analyze the differences between groups using the SPSS 23.0 software, and p-values < 0.05 were considered statistically significant.

## Figures and Tables

**Figure 1 ijms-21-02833-f001:**
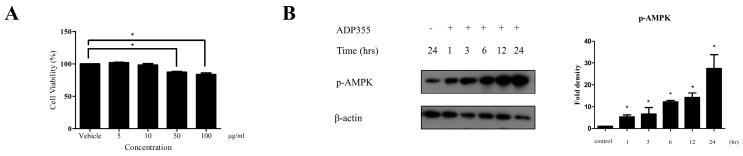
Viability of keloid fibroblasts was evaluated at adiponectin-based peptide (ADP355) doses ranging from 5 to 100 µg/mL. ADP355 affected cell viability significantly at doses ≥ 50 µg/mL (**A**, *p* ≤ 0.05, cut-off > 90%). At 10 µg/mL, ADP355 increased the activation of the AMP-activated protein kinase (AMPK) pathway in a time-dependent manner from 1 to 24 h (**B**, * *p* ≤ 0.05). The data are expressed as mean ± SD. Representative data are shown from three independent experiments.

**Figure 2 ijms-21-02833-f002:**
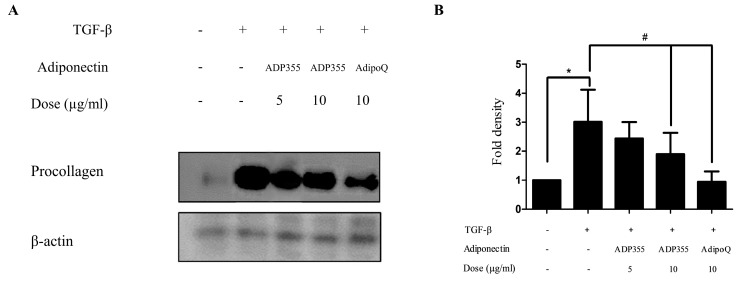
Effect of ADP355 on transforming growth factor β1 (TGF-β1)-induced procollagen expression. Treatment of ADP355 (5 and 10 µg/mL) and adiponectin recombinant, AdipoQ (10 µg/mL), simultaneously with TGF-β1 10 µg/mL of ADP355 and AdipoQ, attenuated TGF-β1-induced procollagen expression. (**A**) Western blot analysis results. (**B**) Quantification of western blot results (* *p* ≤ 0.05, control vs. TGF-β1, # *p* ≤ 0.05, TGF-β1 vs. TGF-β1 + ADP355, and TGF-β1 vs. TGF-β1 + AdipoQ). The data are expressed as mean ± SD. Representative data are shown from three independent experiments.

**Figure 3 ijms-21-02833-f003:**
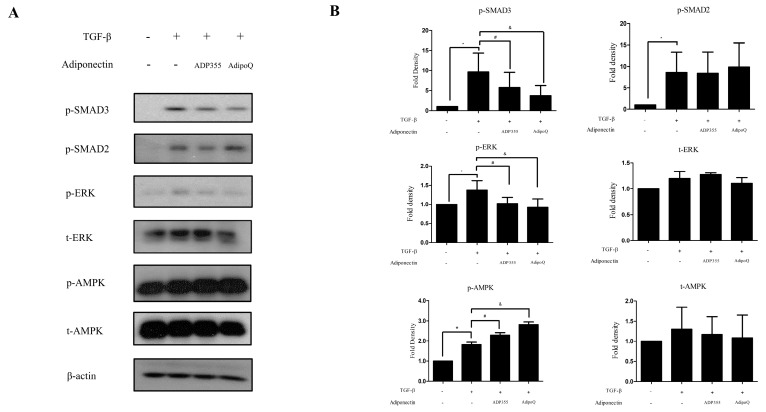
Effect of ADP355 on the TGF-β1-induced downstream pathways. TGF-β1-induced increases in phosphorylation of SMAD2, SMAD3, ERK, and AMPK. Treatment with ADP355 (10 µg/mL) and adiponectin recombinant, AdipoQ (10 µg/mL), reversed TGF-β1-induced phosphorylation of SMAD3 and ERK and accentuated phosphorylation of AMPK. (**A**) Western blot analysis results. (**B**) Quantification of western blot results (* *p* ≤ 0.05, control vs. TGF- β, # *p* ≤ 0.05, TGF- β vs. TGF-β + ADP355, and & *p* ≤ 0.05, TGF-β vs. TGF-β + AdipoQ). The data are expressed as mean ± SD. Representative data are shown from three independent experiments.

**Figure 4 ijms-21-02833-f004:**
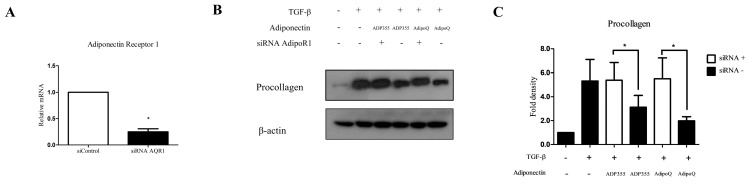
Knockdown of adiponectin receptor 1 (AdipoR1) reversed the ADP355 attenuation of TGF-β1-induced collagen expression. The effect of adiponectin receptor knockdown was compared with that of a negative control siRNA (**A**, * *p* ≤ 0.05). Knockdown of AdipoR1 reversed the procollagen expression attenuation by ADP355 and adiponectin recombinant, AdipoQ (**B**,**C**, * *p* ≤ 0.05). The data are expressed as mean ± SD. Representative data are shown from three independent experiments.

**Figure 5 ijms-21-02833-f005:**
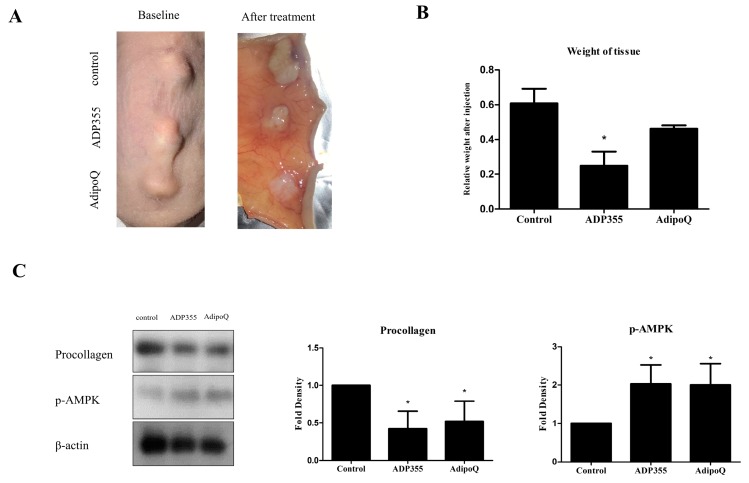
Macrographic examination of xenotransplanted tissue on the back of athymic nude mice before and after treatment (**A**). Weight of xenografted keloid tissues after intralesional injection of ADP355 and AdipoQ. The weight of the tissues was normalized to the baseline weight, and the weight of adiponectin-treated tissue was compared to that of control. ADP355-treated lesions showed a significant weight reduction (**B**, * *p* ≤ 0.05). Procollagen protein expression was significantly reduced while p-AMPK expression increased following treatment with ADP355 and AdipoQ (**C**, * *p* ≤ 0.05). The data are expressed as mean ± SD. Representative data are shown from three independent experiments.

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
