# Peer review of "Adiponectin-Based Peptide (ADP355) Inhibits Transforming Growth Factor-β1-Induced Fibrosis in Keloids"

_ijms, 2020, doi:10.3390/ijms21082833_

Round 1
Reviewer 1 Report
The manuscript is devoted to investigation of adiponectin-based peptide activity towards prevention of keloid scars. It is well designed and written.
I have some minor concern regarding Introduction section.
It seems that Introduction should be more detailed concerning adiponectin signalling pathways and biological effects.
Author Response
Response to Reviewer 1 Comments
General comment: The manuscript is devoted to investigation of adiponectin-based peptide activity towards prevention of keloid scars. It is well designed and written.
Response: We would like to thank you for your careful and thorough reading of the manuscript. We also appreciate your valuable suggestions to improve the quality of this manuscript. As we had our manuscript proofread by professional editing company, there are additional minor changes in English. Please, note that we reduced the abstract to meet the author guideline of IJMS (200 words maximum). Any changes in the manuscript and our response are marked in blue.
Point 1: I have some minor concern regarding Introduction section. It seems that Introduction should be more detailed concerning adiponectin signalling pathways and biological effects.
Response 1: We appreciate your comment. As suggested by the reviewer, we have added additional information in the introduction section as follows (Page 1, Lines 43-49): Adiponectin elicits several downstream signaling events. APPL1 (adaptor protein containing pleckstrin homology domain, phosphotyrosine binding domain, and leucine zipper motif) mediates the intracellular signal transduction of the adiponectin receptor pathway. It activates AMP activated protein kinases (AMPK), PPAR-α, calcium/calmodulin-dependent protein kinase kinase (CaMKK-β), and p38 MAPK. [7, 8]. Adiponectin has distinct target tissue specificity and modulates unique biological processes; it improves insulin sensitivity, regulates energy metabolism, and modulates the inflammatory response [9].
We also added new references:
- Yamauchi, T.; Kamon, J.; Minokoshi, Y.; Ito, Y.; Waki, H.; Uchida, S.; Yamashita, S.; Noda, M.; Kita, S.; Ueki, K.; Eto, K.; Akanuma, Y.; Froguel, P.; Foufelle, F.; Ferre, P.; Carling, D.; Kimura, S.; Nagai, R.; Kahn, B. B.; Kadowaki, T. Adiponectin Stimulates Glucose Utilization and Fatty-Acid Oxidation by Activating AMP-Activated Protein Kinase. Nat. Med. 2002, 8 (11), 1288–1295.
- Deepa, S. S.; Zhou, L.; Ryu, J.; Wang, C.; Mao, X.; Li, C.; Zhang, N.; Musi, N.; DeFronzo, R. A.; Liu, F.; Dong, L. Q. APPL1 Mediates Adiponectin-Induced LKB1 Cytosolic Localization Through the PP2A-PKCzeta Signaling Pathway. Mol. Endocrinol. 2011, 25 (10), 1773–1785.
- Achari, A. E.; Jain, S. K. Adiponectin, a Therapeutic Target for Obesity, Diabetes, and Endothelial Dysfunction. Int. J. Mol. Sci. 2017, 18 (6), 1321.
Reviewer 2 Report
This manuscript looks at ADP355, a shorter peptide derivative of adiponectin, and its ability to inhibit TGFB1-mediated fibrosis is keloid fibroblasts. Though overall the manuscript has merit, there are several issues throughout the manuscript that need to be addressed:
Major
--Materials and Methods: It is unclear for the experiments performed if they were done in duplicate, triplicate, or single well per treatment arm. This is separate from the experiments being done at least three different times, which is stated in the Methods section.
--Figure 1A: Unclear what concentration of ADP355 was used for this experiment; the authors go on to show that higher concentrations of ADP355 can affect cell viability.
--Figure 1A: 6 hour timepoint has a negative sign. Why was ADP355 given at 3 and 12 hours but not as 6 hours? Can the authors explain the increased p-AMPK at 6 hours without ADP355? Or is this a typographical error?
--The Western blots for p-AMPK and t-AMPK do not seen congruent with the graphs in B. In particular, the Westerns do not seem to show a 3 fold upregulation of p-AMPK with AdipoQ compared to control, though this is what is presented in the accompanying bar chart.
--Figure 4B shows no detectable expression of procollagen in control. Figure 4B also shows detectable procollagen when siRNA AdipoR1 is added. However, in 4C, the fold density of control is comparable to the fold density of the two siRNA AdipoR1 knockdowns.
Minor:
--Materials and Methods: Were the keloid samples, that were taken from surgery, naive to prior treatments (eg. intralesional steroids)? Overall, how many samples were taken??
--Materials and Methods, it is not stated what secondary antibodies were used for the detection of p-AMPK and t-AMPK.
Round 2
Reviewer 2 Report
Tha authors have addressed my major and minor concerns with the manuscript. Thank you for addressing all of my concerns so systematically.
My only suggestion is for the abstract (lines 22-23) to change from "...in a primary culture of keloid fibroblasts prepared from..." to "...in primary cultures of keloid fibroblasts prepared from..."